# Multiscale Bayes Adaptive Threshold Wavelet Transform Geomagnetic Basemap Denoising Taking Residual Constraints into Account

**DOI:** 10.3390/s24123847

**Published:** 2024-06-14

**Authors:** Pan Xiong, Gang Bian, Qiang Liu, Shaohua Jin, Xiaodong Yin

**Affiliations:** Department of Military Oceanography and Hydrography and Cartography, Dalian Naval Academy, Dalian 116018, China; zeros0411@163.com (P.X.); liuqiang3931@163.com (Q.L.); jsh_1978@163.com (S.J.); triest@163.com (X.Y.)

**Keywords:** multiscale wavelet transform, geomagnetic reference map, adaptive thresholding, Bayesian estimation, residuals

## Abstract

To achieve high-precision geomagnetic matching navigation, a reliable geomagnetic anomaly basemap is essential. However, the accuracy of the geomagnetic anomaly basemap is often compromised by noise data that are inherent in the process of data acquisition and integration of multiple data sources. In order to address this challenge, a denoising approach utilizing an improved multiscale wavelet transform is proposed. The denoising process involves the iterative multiscale wavelet transform, which leverages the structural characteristics of the geomagnetic anomaly basemap to extract statistical information on model residuals. This information serves as the a priori knowledge for determining the Bayes estimation threshold necessary for obtaining an optimal wavelet threshold. Additionally, the entropy method is employed to integrate three commonly used evaluation indexes—the signal-to-noise ratio, root mean square (RMS), and smoothing degree. A fusion model of soft and hard threshold functions is devised to mitigate the inherent drawbacks of a single threshold function. During denoising, the Elastic Net regular term is introduced to enhance the accuracy and stability of the denoising results. To validate the proposed method, denoising experiments are conducted using simulation data from a sphere magnetic anomaly model and measured data from a Pacific Ocean sea area. The denoising performance of the proposed method is compared with Gaussian filter, mean filter, and soft and hard threshold wavelet transform algorithms. The experimental results, both for the simulated and measured data, demonstrate that the proposed method excels in denoising effectiveness; maintaining high accuracy; preserving image details while effectively removing noise; and optimizing the signal-to-noise ratio, structural similarity, root mean square error, and smoothing degree of the denoised image.

## 1. Introduction

Due to the increasingly complex electromagnetic confrontation environment in modern warfare, satellite navigation is vulnerable to interference and damage during wartime [1]. As a result, there is a growing focus on geomagnetic matching navigation as a means to enhance the anti-jamming capabilities of weapon platform navigation. Geomagnetic matching navigation is gaining prominence because of the stability of the geomagnetic field, its immunity to the geographic location and weather conditions, and its classification as a passive detection signal with significant concealment properties, making it a valuable auxiliary navigation tool in wartime scenarios. This approach involves the utilization of real-time magnetic anomaly data and an established geomagnetic anomaly reference map for precise positioning, necessitating the highest level of accuracy in the geomagnetic anomaly reference map [2].

To create the geomagnetic anomaly reference map, geomagnetic anomaly data from the region are gathered and compiled into a map using shipborne or aeromagnetic measurements. Throughout this process, several corrections are essential, including adjustments for daily variation, ship magnetic effects, position normalization, and instrument errors. However, even after applying these corrections, residual noise resulting from these errors tends to persist, making complete eradication challenging. Given that the compilation of the geomagnetic anomaly reference map involves amalgamating magnetic data from various time periods, instruments, and voyages, the introduction of noise is inevitable [3]. This integration of data from different sources necessitates the filtering of noise in the geomagnetic anomaly reference map to enhance the accuracy of geomagnetic matching navigation.

At present, commonly used denoising methods in measurement data processing include Gaussian filtering, mean value filtering, and soft and hard threshold wavelet denoising methods. Gaussian filtering is a commonly used image denoising technique with the advantages of a good smoothing effect and simple calculation, but its filtering process is also a fuzzy process of image processing, which will cause the loss of the original image edge information [4]. Mean value filtering has the advantages of being simple and easy to implement and better preservation of the overall color tone and brightness of the image, but it is unable to effectively remove large outliers that cause artifacts in the image; because mean value filtering is essentially a linear smoothing filter, the edges of the feature region and texture information will be lost when filtering [5]. Wavelet transform was firstly proposed by Mallat et al. in 1989, and after decades of development, it has become one of the most important methods for denoising all kinds of data; wavelet transform has the advantages of strong tunability, multiscale processing, time-frequency localization, etc. [6]. However, wavelet denoising involves many positive transforms and inverse transforms, which makes the computational quantity larger, and the size of the threshold, the setting of the threshold function, and the selection of wavelet bases has a greater impact on the experimental results. Therefore, the appropriate threshold, threshold function, and wavelet base are the key to the denoising effect [7]. The design of soft and hard threshold functions has also become a hot research topic; the soft threshold function focuses on smoothing the image and retaining the image details, while the hard threshold function can remove the noise more thoroughly, but it will lose some detail information, resulting in a loss of accuracy [8]. Researchers have spent a lot of effort to integrate the respective advantages of the soft threshold function and the hard threshold function [9] to obtain a moderate threshold function. And whether it is a soft threshold function or a hard threshold function, their denoising effect is dependent on setting the appropriate size of the threshold, and the selection of the threshold is very critical for the denoising effect [10]. The threshold can be determined by the minimum mean square error criterion to minimize the overall distortion of the noise and the signal and also by analyzing the statistical properties of the noise-containing data to distinguish the wavelet coefficients from the noise to determine the appropriate threshold. The Bayesian estimation threshold principle is a signal processing method based on Bayesian theory, which is used to determine the appropriate signal threshold in the process of wavelet transform denoising, and its basic idea is to determine the optimal threshold by maximizing the a posteriori probability or minimizing the expected loss [11]. However, the Bayesian determination of the threshold requires estimation of the probability distributions of the signal and noise, but in general, this part of the a priori information is more difficult to obtain.

In this paper, we propose to use the residual information as the a priori information of the Bayesian estimation threshold for the iterative derivation of the threshold and put forward a fusion model of soft and hard threshold functions to take into account the advantages of each of the soft and hard threshold functions. Considering that the geomagnetic reference map has higher requirements for detail information retention, a regularization term is introduced in the selection of wavelet coefficients to improve the accuracy of the filtering, and the superiority of the algorithm in this paper compared with the traditional algorithms is verified by the simulation data and the measured data in this paper.

## 2. Model Building

### 2.1. Bayesian Threshold Modeling with Residuals in Mind

The key to the denoising effect of multiscale wavelet transform lies in the appropriate selection of the wavelet bases, the calculation of the optimal threshold, and the determination of the optimal decomposition scale, while the a priori distribution information of the parameters needs to be obtained in the Bayesian estimation [12]. Therefore, in this paper, in view of the noise characteristics of the magnetic reference map, the a priori information of the noise distribution is determined by designing the cyclic residual model, and the optimal information in the wavelet decomposition is further projected.

Firstly, the original Bayesian threshold wavelet transform is used to obtain the residual information, and the obtained residuals are used to calculate their statistical distribution information, which is used as the prior distribution of the Bayesian estimation and to estimate the thresholds, and the obtained new thresholds are used as the update parameters to carry out the multiscale wavelet transform again. The peak signal-to-noise ratio can represent the quality of the denoised image, and when the peak signal-to-noise ratio of the resulting image reaches the inflection point, it is recognized as the optimal threshold under the current wavelet basis and decomposition scales, while the wavelet basis and decomposition scales are alternately optimized by constructing the binary tree model and using the depth-first search until the convergence condition is reached. On the basis of the optimal parameters computed by the model, a regularization term is introduced to limit the influence of the denoising process on the magnetic image and to improve the denoising accuracy. The model can calculate the optimal wavelet basis, decomposition scale, and threshold value adapted to the noise of the target geomagnetic reference map and ensure the reliability and accuracy of the data quality on the basis of denoising. The specific formulas which take into account the residual Bayesian wavelet threshold are as follows:

Let the mean of the residuals be μ, the variance is $\sigma_{S}^{2}$, and the formula is as follows:(1)$$p(S)=(1/\sqrt{2\pi }\sigma_{S})\exp(-S^{2}/2\sigma_{S}^{2})$$
where p(S) denotes the information about the statistical distribution of the residual S. Let X be the original image with distribution $GGD_{\left(\alpha_{X},\beta \right)}$, $\sigma_{X}$ is the standardized variance of X, and β is the shape parameter, which here takes the value 2.

By calculating the resulting prior information, find the threshold that minimizes the Bayesian risk T, let $\overset{\Lambda }{X}=\eta_{t}(y)$, $y/X\sim N(X,{\sigma_{S}}^{2})$, and the Bayesian risk is expressed as follows:(2)$$r(T)=E{\left(\overset{\Lambda }{X}-X\right)}^{2}=E_{x}E_{y|x}{\left(\overset{\Lambda }{X}-X\right)}^{2}$$
(3)$$\begin{array}{l} E_{X}E_{Y|X}(\overset{\Lambda }{X}-X)=\int_{-\infty }^{+\infty }\int_{-\infty }^{+\infty }{\left(\eta_{t}(y)-x\right)}^{2}p(y|x)p(x)d_{y}d_{x}\\ ={\sigma_{S}}^{2}\omega (\sigma_{X}^{2}/{\sigma_{S}}^{2},T/\sigma_{S}) \end{array}$$
(4)$$\omega (\sigma_{X}^{2},T)=\sigma_{X}^{2}+2(T^{2}+1-\sigma_{X}^{2})\bar{\phi }(T/\sqrt{1+\sigma_{X}^{2}})-2T(1+\sigma_{X}^{2})\phi (T,1+\sigma_{X}^{2})$$

The probability density function is:(5)$$\phi (x,{\sigma_{S}}^{2})=(1/\sqrt{2\pi {\sigma_{S}}^{2}})\exp(-(x^{2}/2{\sigma_{S}}^{2}))$$
(6)$$\bar{\phi }(x)=\int_{x}^{\infty }\phi (t,1)dt$$

In the above equations, $\phi (x,{\sigma_{S}}^{2})$ is the probability density function, $\overset{\Lambda }{X}$ is the estimate of X, Y is the observed image, r(T) denotes the Bayesian risk, $E{\left(\overset{\Lambda }{X}-X\right)}^{2}$ is the error risk function, and the image denoising is performed to obtain an estimate Y of X using image $\overset{\Lambda }{X}$. To minimize the error risk function, it can be obtained:(7)$$T^{*}=\arg\;\underset{T}{\min}\;r(T)=\frac{\sigma_{S}^{2}}{\sigma_{X}}$$

Equation (7) minimized the Bayesian risk function introduced by Equations (2)–(6).
(8)$$\sigma_{Y}^{2}=\frac{1}{n^{2}}\sum_{s,q=1}^{m}Y_{sq}^{2}$$
where $Y_{sq}$ is the subband coefficient of the wavelet transform.
(9)$$\sigma_{X}=\sqrt{\max(\sigma_{Y}^{2}-\sigma_{S},0)}$$

The threshold for each iteration is calculated as:(10)$$T={\sigma_{S}}^{2}/\sigma_{X}$$

The optimal threshold is calculated by bringing Equations (8) and (9) into Equation (10).

In order to improve the accuracy and stability of the denoising results, Elastic Net regularization is used to optimize the wavelet coefficients during the wavelet transform, and the regularization formula is as follows:(11)$$\min{\left(X-L\times C_{i}\right)}^{2}+\lambda_{1}\times |C_{i}|+\lambda_{1}\times {C_{i}}^{2}$$
where L is the wavelet transform matrix, $C_{i}$ denotes the wavelet coefficients of the ith rewavelet transform, and $\lambda_{1}$ and $\lambda_{2}$ are the sparse regularization and ridge regression parameters, respectively.

The flowchart of the improved multiscale variational wavelet transform algorithm is shown below (see Figure 1).

### 2.2. Soft and Hard Threshold Fusion Function Models

Soft threshold function denoising will make the image tend to be smooth, does not produce ringing, have a pseudo-Gibbs effect, and at the same time can better retain the details of the signal and can provide a higher signal-to-noise ratio. But with the increase in the number of layers of decomposition, it will make the image’s boundaries blurred, with a distortion phenomenon, and the computational complexity is higher. Hard threshold function denoising can retain the sharp changes in the signal and can effectively filter out low-amplitude noise, and the calculation speed is faster, but in the case of a low signal-to-noise ratio, it is prone to signal loss and will produce artifacts. Therefore, in order to synthesize the advantages of the two proposed soft and hard threshold function fusion models, the signal-to-noise ratio, root mean square, and smoothness of the denoised image, respectively, are calculated using the entropy method for the fusion of the indicators [13]. The weighted fusion of the soft and hard threshold function calculations are determined using the calculated fusion metrics [14]. The specific expression is:(12)$$F(m)=\omega_{vrm}C_{vrm}(m)+\omega_{vsnr}C_{vsnr}(m)+\omega_{vs}C_{vs}(m)$$
where F(m) is the calculated integration indicator; $\omega_{vrm}$, $\omega_{vsnr}$, $\omega_{vs}$ are the weights of the root mean square error, the signal-to-noise ratio, and the change in smoothness obtained from the entropy calculation, respectively; m denotes the wavelet decomposition scale; $C_{vrm}$, $C_{snr}$, $C_{vs}$ denote the normalization result of each quantity, respectively, and the expression is:(13)$$C_{vrm}(m)=[vrm(m)-vrm_{\min}]/(vrm_{\max}-vrm_{\min})$$
(14)$$C_{vsnr}(m)=[vsnr(m)-vsnr_{\min}]/(vsnr_{\max}-vsnr_{\min})$$
(15)$$C_{vs}(m)=[vs(m)-vs_{\min}]/(vs_{\max}-vs_{\min})$$

The amount of change in the signal-to-noise ratio error is given as an example of how the weights of the indicators are calculated as:(16)$$\omega_{C_{vsnr}}=\frac{1-H_{C_{vsnr}}}{\left(1-H_{C_{vsnr}})+(1-H_{C_{vrm}})+(1-H_{C_{vs}}\right)}$$
(17)$$H_{C_{vsnr}}=-(\frac{1}{\ln(N)})\sum_{i=1}^{N}P_{i}^{C_{vsnr}}\ln P_{i}^{C_{vsnr}}$$
(18)$$P_{i}^{C_{vsnr}}=C_{vsnr}(i)/\sum_{i=1}^{N}C_{vsnr}(i)$$

And the limit difference threshold is set. When the difference between the fusion indexes of the denoised images of different function models is less than the threshold, the model is combined by calculating the fusion weights of each index, and when the difference is greater than the threshold, the fusion is not carried out and the optimal model is used directly. The realization process is as follows:(19)$$W_{hard}=\frac{F{\left(m\right)}_{hard}}{F{\left(m\right)}_{hard}+F{\left(m\right)}_{soft}}$$
(20)$$W_{soft}=\frac{F{\left(m\right)}_{soft}}{F{\left(m\right)}_{hard}+F{\left(m\right)}_{soft}}$$
where $W_{hard}$, $W_{soft}$ are the fusion model weights of the soft and hard threshold functions, and $F{\left(m\right)}_{soft}$, $F{\left(m\right)}_{hard}$ are the fusion metrics computed by the soft and hard threshold functions, respectively.
(21)$$\frac{\left|F{\left(m\right)}^{hard}-F{\left(m\right)}^{soft}\right|}{\max(F{\left(m\right)}^{hard}-F{\left(m\right)}^{soft})}>\theta$$

In order to improve the generalization ability of the model, by setting a threshold value θ, when the fusion index calculated by the soft and hard threshold functions satisfies the above equation, the model fusion is not carried out, and the model with the large fusion index of the two is directly selected.

## 3. Test and Analysis

### 3.1. Assessment Method

In order to verify the effectiveness of the proposed multimodal Bayes adaptive threshold wavelet geomagnetic basemap denoising algorithm considering residuals, the proposed algorithm is compared and analyzed by using mean filtering, Gaussian filtering, soft threshold wavelet filtering, and hard threshold wavelet filtering; in order to evaluate the performance of different denoising methods, four evaluation metrics, namely, the peak signal-to-noise ratio (PSNR), structural similarity index (SSIM), root mean square error (RMSE), and smoothness, are used to analyze and discuss the experiments. 

The PSNR can be used to evaluate the degree of distortion of the denoised image; the larger the PSNR, the better the quality of the denoised image [15]. The PSNR is calculated using the following formula:(22)$$PSNR=10\times \log_{10}\left(\frac{N\times MAX_{I}^{2}}{\sum_{i=1}^{N}\left(I_{orig(i)}-I_{comp{\left(i\right)}^{2}}\right)}\right)$$
where $MAX_{I}^{2}$ represents the maximum value of the basemap, $I_{orig(i)}$ is the value of the original base image, and $I_{comp}$ represents the value of the processed image.

The structural similarity index is used to evaluate the denoising quality of an image by comparing the similarity, brightness, and structure between images, which can be used to compare the difference between geomagnetic datum maps. The larger value of the SSIM represents the higher quality of the image denoising. The formula for calculating the SSIM can be expressed as:(23)$$SSIM(x,y)=l{\left(x,y\right)}^{\alpha }c{\left(x,y\right)}^{\beta }s{\left(x,y\right)}^{\gamma }$$
(24)$$SSIM(x,y)=\frac{\left(2\mu_{x}\mu_{y}+C_{1})(2\sigma_{x}\sigma_{y}+C_{2}\right)}{\left(\mu_{x}^{2}+\mu_{y}^{2}+C_{1})(\sigma_{x}^{2}+\sigma_{y}^{2}+C_{2}\right)}$$
where l(x,y) denotes the brightness, c(x,y) denotes the contrast, s(x,y) denotes the structure, x denotes the original image, and y denotes the noisy image.

The accuracy of different denoising algorithms can be reflected by calculating the root mean square error between the denoised image and the original image and examining the effect of denoising algorithms on the accuracy of geomagnetic reference maps [16].
(25)$$RMSE=\sqrt{\frac{\sum_{i=1}^{n}{\left(X_{obs,i}-X_{\mathrm{mod}el,i}\right)}^{2}}{n}}$$
where $X_{obs,i}$ represents the value of the original basemap and $X_{\mathrm{mod}el,i}$ represents the value of the processed basemap.

The smoothness can evaluate the image texture and the degree of change in the magnetic reference map after denoising, and in this paper, the gradient calculation method is used to calculate the smoothness.
(26)$${\left[G_{x},G_{y}\right]}^{T}={\left[\frac{\delta f}{\delta x},\frac{\delta f}{\delta y}\right]}^{T}$$
(27)$$g(x,y)=\sqrt{\frac{\partial^{2}f}{\partial x^{2}}+\frac{\partial^{2}f}{\partial y^{2}}}$$
where G(x), G(y) is the gradient of the image f(x,y) in the x direction and the y direction at the point (x,y), and g(x,y) is the smoothness of the point.

### 3.2. Results and Discussion

#### 3.2.1. Simulation Experiment

The magnetic anomaly basemap generated by the sphere magnetic anomaly model is used for the simulation experiments, and the parameters of the sphere are set as follows: a magnetization rate of 0.15, magnetic field strength of 50 nT, radius of the sphere of 250 m, and burial depth of 100 m. In order to visualize the denoising effect, the Gaussian noise with a variance of 0.03, 0.05, 0.07, and 0.09 is added to the magnetic anomaly reference map generated by the sphere magnetic anomaly model, and the mean filter, Gaussian filter, soft threshold wavelet filter, and hard threshold wavelet filter are used to compare the algorithms with the proposed algorithms, and the denoising results are analyzed to judge the five filtering algorithms on the processing of the noise of the magnetic reference map. The ability of the five filtering algorithms to deal with the noise of the magnetic reference map is analyzed. The experimental environment of this paper is configured as follows: the CPU is an AMD Ryzen 7 4800 H (Advanced Micro Devices, Santa Clara, CA, America), the GPU is an NVIDIA GeForce RTX 2060 (NVIDIA, Santa Clara, CA, America), and the development language is 2023b MATLAB.

In Figure 1, Figure 2, Figure 3, Figure 4 and Figure 5, The purple area in the graph above is a positive magnetic anomaly, which slowly turns to a negative anomaly as it moves closer to the blue color. In Figure 2(g1,g2), Figure 3(g1,g2), Figure 4(g1,g2), and Figure 5(g1,g2), it can be seen that the simulated magnetic anomaly map benchmark map processed by the algorithm proposed in this paper has clear boundaries and high smoothness, and the denoising effect is better than the remaining four algorithms.

In order to better observe the denoising effect of the different algorithms in the magnetic anomaly simulation datum map under different noise levels, the changes in four commonly used denoising judgments, the PSNR, SSIM, RMSE, and smoothness, are used to compare the denoising ability of each algorithm [17]. As seen in Figure 6, Figure 7, Figure 8 and Figure 9, the algorithm proposed in this paper is better than the remaining four algorithms in all the indicators, especially in the indicator of smoothness where it is far better than the rest of the denoising algorithms. In Figure 6, the curve of this paper’s model decreases the curvature of the PSNR most slowly with the increase in noise, and it can be seen that it is less affected by noise. In Figure 7, the RMSE of this paper’s model increases most slowly with the increase in noise, and it can also be concluded that this paper’s model is less affected by noise and is more robust. In Figure 9, the similarity of this paper’s model decreases by only 0.06% with the increase in noise, which is the lowest degree of decrease compared to the rest of the algorithms.

With the magnetic anomaly reference map in the geomagnetic matching navigation for the data of the higher accuracy requirements and the root mean square error that can be a better response to the different denoising algorithms for the accuracy of the original data to maintain the level of Figure 6, it can be seen that after using the algorithm of this paper that denoising the calculation of the root mean square error is the lowest, and the denoised data can be used to calculate the accuracy of the data of the original data. The root mean square error is the lowest and the accuracy of the denoised image is the highest.

#### 3.2.2. Measured Data Analysis and Validation

In order to further test the denoising effect of this paper’s algorithm in the actual situation, the measured geomagnetic anomaly benchmark map of a sea area in the Pacific Ocean is selected for the denoising comparison experiments; the size of the region is 110 km × 110 km, the spacing of the measurement lines is 1 km, and the original magnetic anomaly map is shown in Figure 5(a1). And in order to more intuitively illustrate the denoising effect, this paper uses the artificial inclusion of the variance of 0.05, 0.08, 0.11, and 0.14 Gaussian noise in the original data; the same mean filtering, Gaussian filtering, soft threshold wavelet filtering, and hard threshold wavelet filtering; and the proposed algorithm for the comparison experiments, analyzing the denoising results to judge the ability of the five filtering algorithms to deal with the noise of the magnetic reference map.

As can be seen from Figure 10, Figure 11, Figure 12 and Figure 13, the denoising effect of this paper’s algorithm is better than the rest of the algorithms in the measured magnetic anomaly benchmark map under different noise levels. After the denoising of this paper’s algorithm, the edge information of the benchmark map is better retained, and the edge information of the magnetic anomaly features is clearer, which can be observed intuitively in the advantages of this paper’s algorithm, both in the true-color and grayscale images. In Figure 10(g1,g2), Figure 11(g1,g2), Figure 12(g1,g2) and Figure 13(g1,g2), it can be seen that this paper’s algorithm basically does not lose the details of the reference image while removing the noise, and it has the highest retention, while the other methods lose a small amount of detail and appear distorted.

In Figure 10, Figure 11, Figure 12 and Figure 13, The purple area in the graph above is a positive magnetic anomaly, which slowly turns to a negative anomaly as it moves closer to the blue color. In order to objectively analyze the denoising effect, in addition to the visual effect being better than the remaining four algorithms, the denoising effect is also evaluated by calculating the PSNR, SSIM, RMSE, and smoothness of the denoised image, and the experimental results are shown in Figure 14, Figure 15, Figure 16 and Figure 17. As seen in Figure 10 and Figure 11, this paper’s algorithm has a lower level of smoothness on the image and maintains the highest level of accuracy, indicating that this paper’s algorithm loses less detail information on the geomagnetic datum map, and the rest of the comparative algorithms denoise the geomagnetic datum map excessively, removing the detail information as noisy data, resulting in a lower level of accuracy and a higher level of smoothness.

Similarly, in order to evaluate the accuracy level of each algorithm in maintaining the original measured data, the size of the root mean square error of the denoising results of different algorithms can be seen in Figure 15. The algorithm of this paper has the smallest root mean square error, and with the increase in the noise, the root mean square error rate of change is smaller, and its performance is better for the robustness of the noise. The SSIM reflects the structural similarity between the denoised image and the original image, and the higher the SSIM, the better the visual effect of the denoised image is. The SSIM can reflect the structural similarity between the denoised image and the original image; the higher the SSIM, the better the visual effect after denoising the benchmark image, which can also reflect the effect of the denoising algorithm in retaining the details of the benchmark image. In Figure 17, it can be seen that the algorithm in this paper, in the case of the increasing noise level, results in the SSIM decreasing more slowly than the rest of the algorithms, and the denoising effect is less affected by the noise level.

## 4. Conclusions

This paper focuses on the denoising method of a geomagnetic anomaly basemap and proposes a multimodal Bayes adaptive threshold wavelet geomagnetic basemap denoising method taking residuals into account. It is shown by the simulation experiments and measured data that the denoising effect of this method is better than that of mean value filtering, Gaussian filtering, soft-thresholding wavelet filtering, and hard-thresholding wavelet filtering under the same conditions. In addition, the method limits the modification of the original measurements by the filtering algorithm by setting the inclusion of the regularization term, which effectively reduces the root mean square error, which is conducive to the improvement in the accuracy of the reference map in geomagnetic matching navigation. Through simulation experiments and real data verification, it can be seen that this paper’s algorithm, in the case of the increasing noise level, results in the denoising effect decreasing more slowly, and the robustness to noise is better. This method can play an important role in practical application scenarios, such as geomagnetic data denoising, the preparation of geomagnetic reference maps, and geomagnetic matching navigation. In the next study, we will continue to optimize the computational efficiency of the algorithm, deeply explore the denoising effect of different wavelet bases in the context of a geomagnetic field, and use the optimization algorithm to further improve the denoising efficiency of the algorithm.

## Figures and Tables

**Figure 1 sensors-24-03847-f001:**
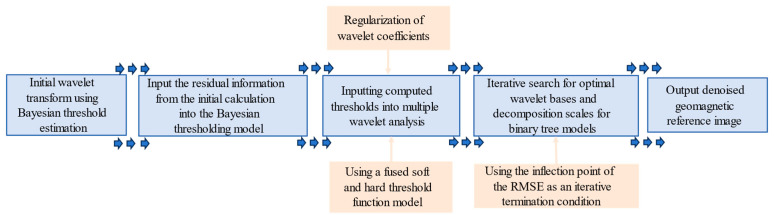
Flowchart of the improved multiscale wavelet transform algorithm.

**Figure 2 sensors-24-03847-f002:**
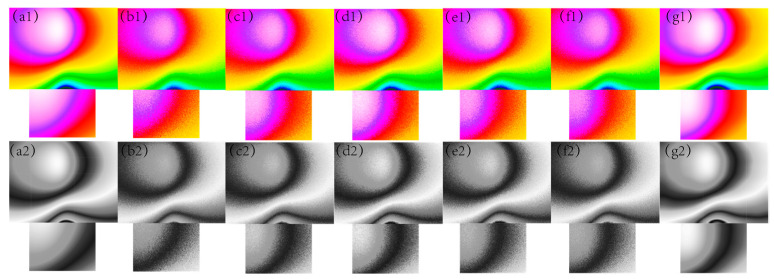
The filtering effect of the simulated data of each algorithm when the variance of the Gaussian noise is 0.003, (**a1**) represents the original magnetic anomaly benchmark image, (**b1**) the added noise magnetic anomaly benchmark image, (**c1**) the Gaussian filter denoising effect image, (**d1**) the mean filtering effect image, (**e1**) the soft-thresholding wavelet filtering effect image, (**f1**) the hard-thresholding wavelet filtering effect image, (**g1**) the filtering effect image of this paper’s algorithm, respectively, and (**a2**), (**b2**), (**c2**), (**d2**), (**e2**), (**f2**) and (**g2**) are the corresponding grayscale images.

**Figure 3 sensors-24-03847-f003:**
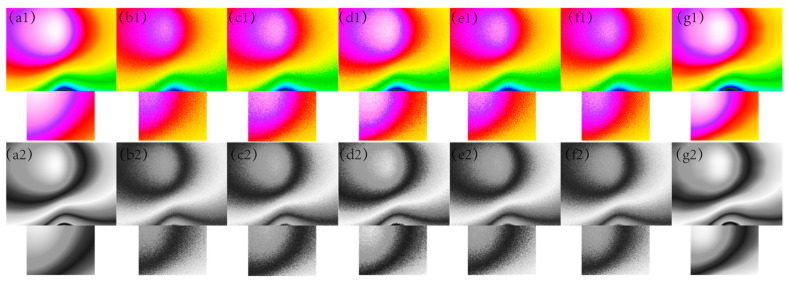
The filtering effect of the simulated data of each algorithm when the variance of the Gaussian noise is 0.005, (**a1**) represents the original magnetic anomaly benchmark image, (**b1**) the added noise magnetic anomaly benchmark image, (**c1**) the Gaussian filter denoising effect image, (**d1**) the mean filtering effect image, (**e1**) the soft-thresholding wavelet filtering effect image, (**f1**) the hard-thresholding wavelet filtering effect image, (**g1**) the filtering effect image of this paper’s algorithm, respectively, and (**a2**), (**b2**), (**c2**), (**d2**), (**e2**), (**f2**) and (**g2**) are the corresponding grayscale images.

**Figure 4 sensors-24-03847-f004:**
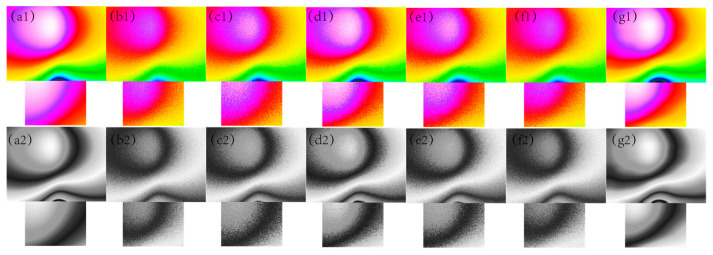
The filtering effect of the simulated data of each algorithm when the variance of the Gaussian noise is 0.007, (**a1**) represents the original magnetic anomaly benchmark image, (**b1**) the added noise magnetic anomaly benchmark image, (**c1**) the Gaussian filter denoising effect image, (**d1**) the mean filtering effect image, (**e1**) the soft-thresholding wavelet filtering effect image, (**f1**) the hard-thresholding wavelet filtering effect image, (**g1**) the filtering effect image of this paper’s algorithm, respectively, and (**a2**), (**b2**), (**c2**), (**d2**), (**e2**), (**f2**) and (**g2**) are the corresponding grayscale images.

**Figure 5 sensors-24-03847-f005:**
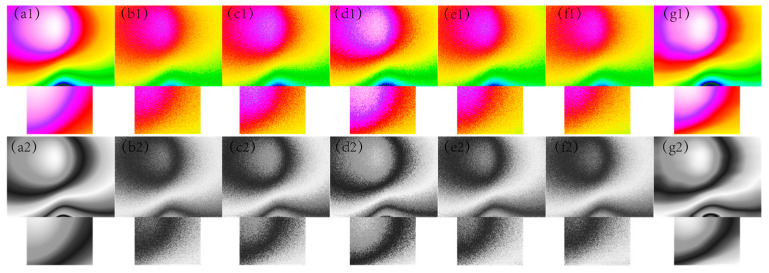
The filtering effect of the simulated data of each algorithm when the variance of the Gaussian noise is 0.009, (**a1**) represents the original magnetic anomaly benchmark image, (**b1**) the added noise magnetic anomaly benchmark image, (**c1**) the Gaussian filter denoising effect image, (**d1**) the mean filtering effect image, (**e1**) the soft-thresholding wavelet filtering effect image, (**f1**) the hard-thresholding wavelet filtering effect image, (**g1**) the filtering effect image of this paper’s algorithm, respectively, and (**a2**), (**b2**), (**c2**), (**d2**), (**e2**), (**f2**), and (**g2**) are the corresponding grayscale images.

**Figure 6 sensors-24-03847-f006:**
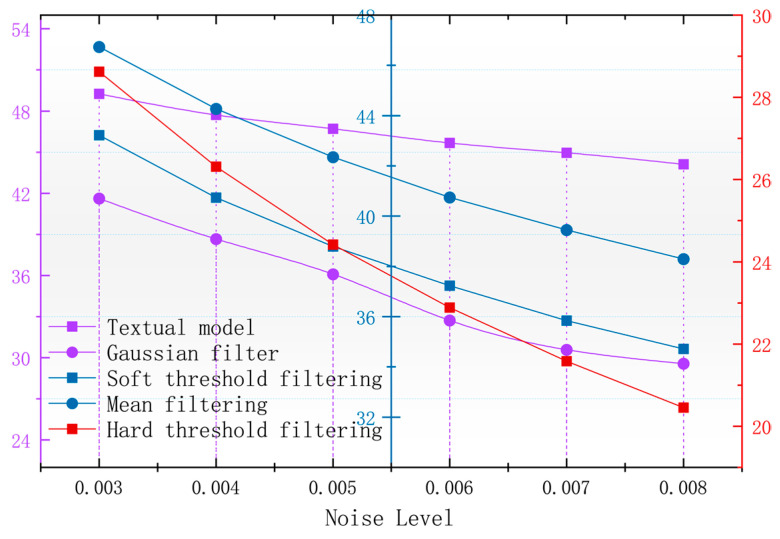
Variation in image PSNR with noise after denoising with different algorithms.

**Figure 7 sensors-24-03847-f007:**
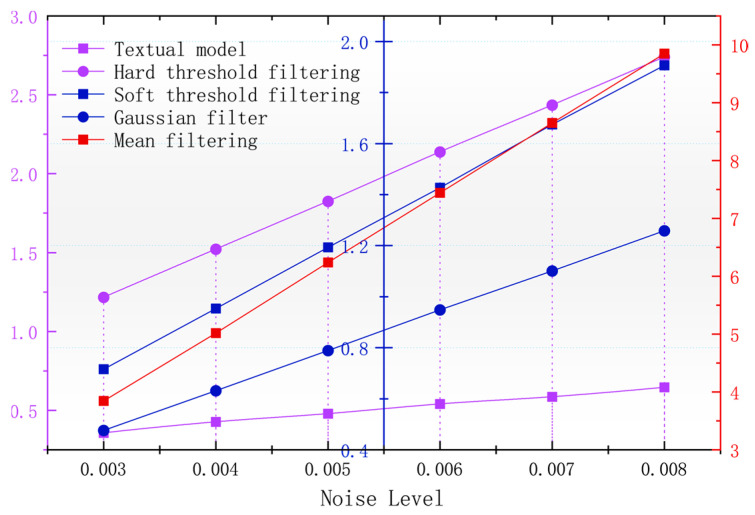
Variation in image RMSE with noise after denoising by different algorithms.

**Figure 8 sensors-24-03847-f008:**
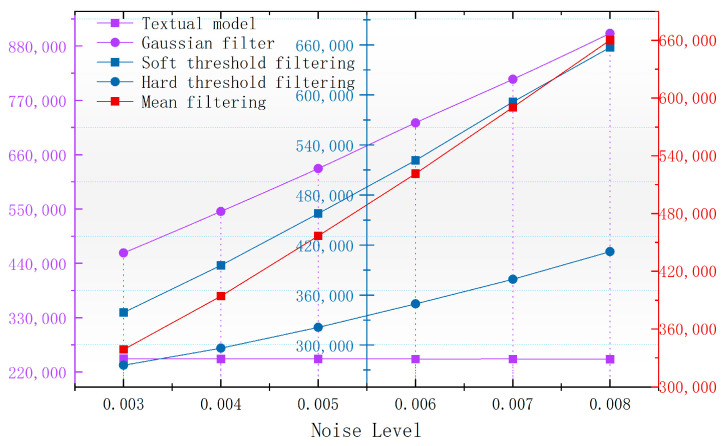
Variation in image smoothness with noise after denoising by different algorithms.

**Figure 9 sensors-24-03847-f009:**
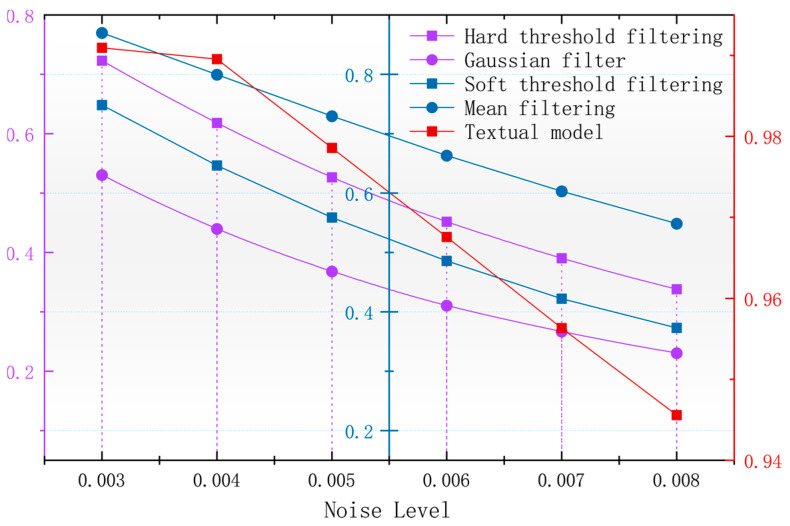
Variation in image SSIM with noise after denoising by different algorithms.

**Figure 10 sensors-24-03847-f010:**
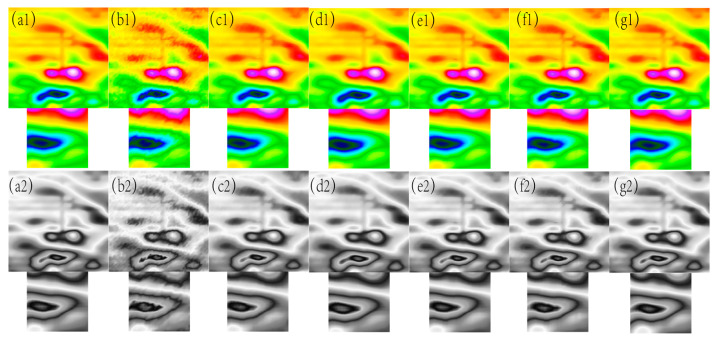
The filtering effect of the measured data of each algorithm when the variance of the Gaussian noise is 0.05, (**a1**) represents the original magnetic anomaly benchmark image, (**b1**) the added noise magnetic anomaly benchmark image, (**c1**) the Gaussian filter denoising effect image, (**d1**) the mean filtering effect image, (**e1**) the soft-thresholding wavelet filtering effect image, (**f1**) the hard-thresholding wavelet filtering effect image, (**g1**) the filtering effect image of this paper’s algorithm, respectively, and (**a2**), (**b2**), (**c2**), (**d2**), (**e2**), (**f2**) and (**g2**) are the corresponding grayscale images.

**Figure 11 sensors-24-03847-f011:**
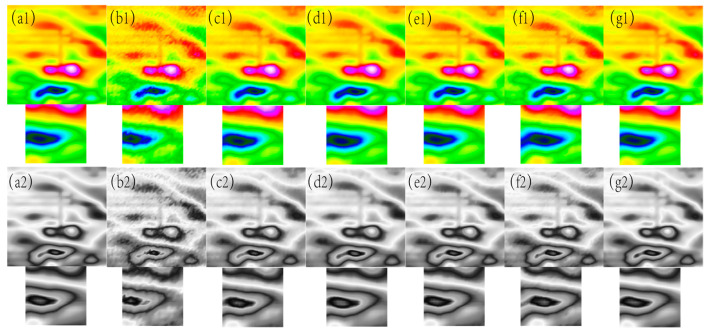
The filtering effect of the measured data of each algorithm when the variance of the Gaussian noise is 0.08, (**a1**) represents the original magnetic anomaly benchmark image, (**b1**) the added noise magnetic anomaly benchmark image, (**c1**) the Gaussian filter denoising effect image, (**d1**) the mean filtering effect image, (**e1**) the soft-thresholding wavelet filtering effect image, (**f1**) the hard-thresholding wavelet filtering effect image, (**g1**) the filtering effect image of this paper’s algorithm, respectively, and (**a2**), (**b2**), (**c2**), (**d2**), (**e2**), (**f2**) and (**g2**) are the corresponding grayscale images.

**Figure 12 sensors-24-03847-f012:**
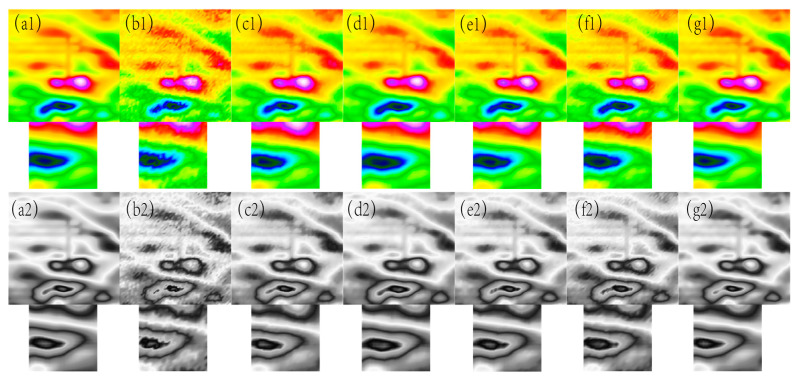
The filtering effect of the measured data of each algorithm when the variance of the Gaussian noise is 0.11, (**a1**) represents the original magnetic anomaly benchmark image, (**b1**) the added noise magnetic anomaly benchmark image, (**c1**) the Gaussian filter denoising effect image, (**d1**) the mean filtering effect image, (**e1**) the soft-thresholding wavelet filtering effect image, (**f1**) the hard-thresholding wavelet filtering effect image, (**g1**) the filtering effect image of this paper’s algorithm, respectively, and (**a2**), (**b2**), (**c2**), (**d2**), (**e2**), (**f2**), and (**g2**) are the corresponding grayscale images.

**Figure 13 sensors-24-03847-f013:**
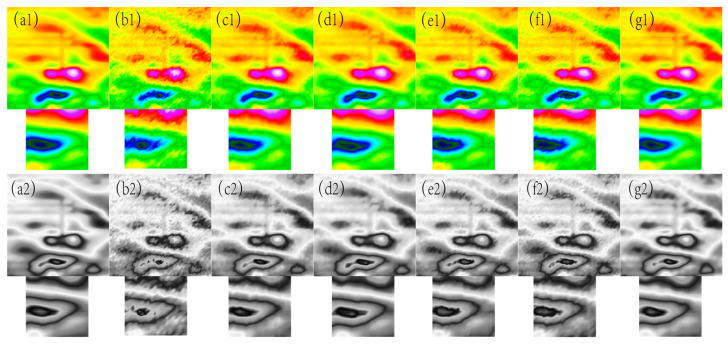
The filtering effect of the measured data of each algorithm when the variance of the Gaussian noise is 0.14, (**a1**) represents the original magnetic anomaly benchmark image, (**b1**) the added noise magnetic anomaly benchmark image, (**c1**) the Gaussian filter denoising effect image, (**d1**) the mean filtering effect image, (**e1**) the soft-thresholding wavelet filtering effect image, (**f1**) the hard-thresholding wavelet filtering effect image, (**g1**) the filtering effect image of this paper’s algorithm, respectively, and (**a2**), (**b2**), (**c2**), (**d2**), (**e2**), (**f2**), and (**g2**) are the corresponding grayscale images.

**Figure 14 sensors-24-03847-f014:**
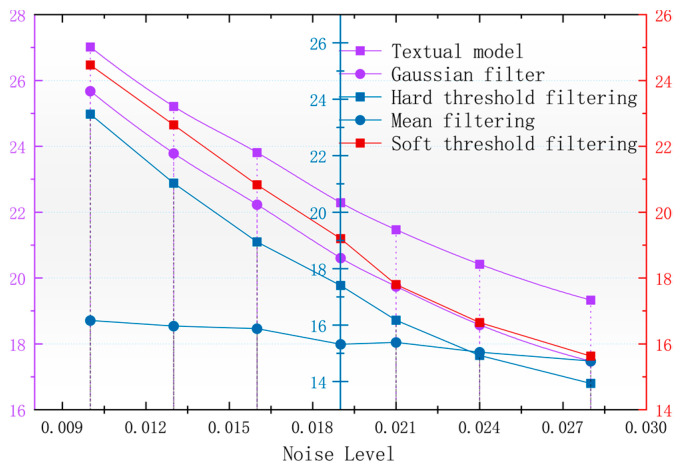
Variation in image PSNR with noise after denoising with different algorithms.

**Figure 15 sensors-24-03847-f015:**
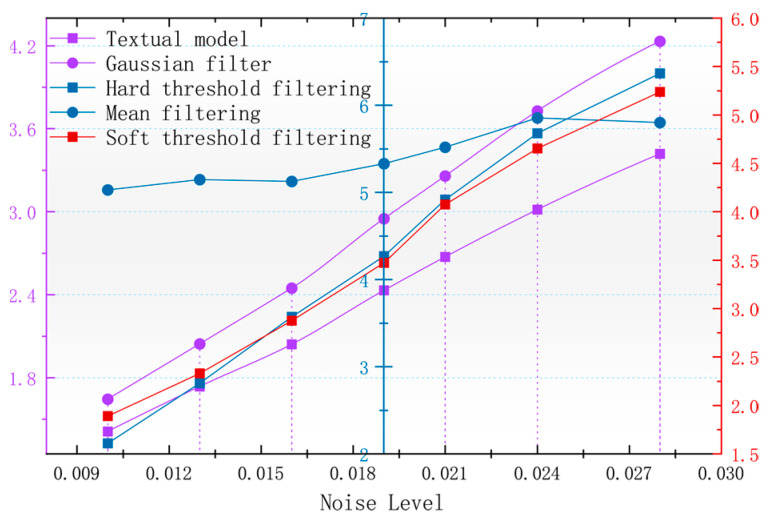
Variation in image RMSE with noise after denoising by different algorithms.

**Figure 16 sensors-24-03847-f016:**
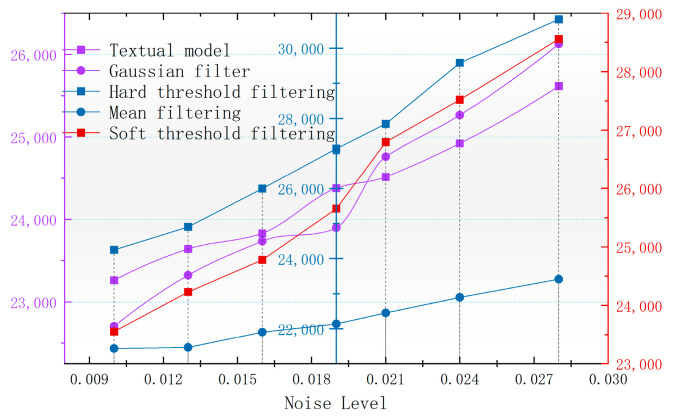
Variation in image smoothness with noise after denoising by different algorithms.

**Figure 17 sensors-24-03847-f017:**
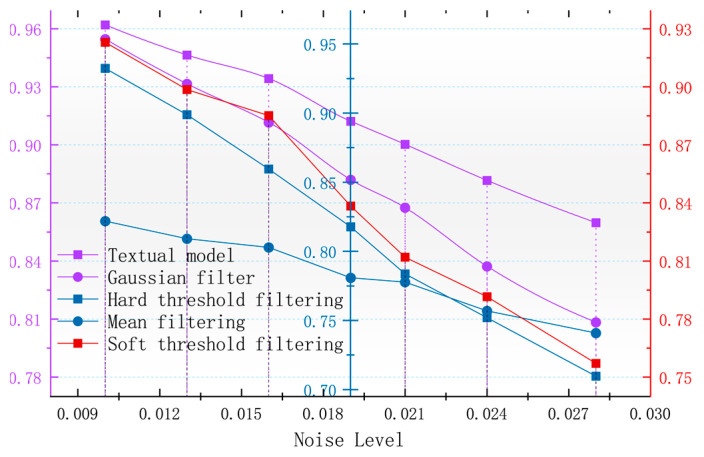
Variation in image SSIM with noise after denoising by different algorithms.

## Data Availability

Data are contained within the article.

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
