# Peer review of "Multiscale Bayes Adaptive Threshold Wavelet Transform Geomagnetic Basemap Denoising Taking Residual Constraints into Account"

_sensors, 2024, doi:10.3390/s24123847_

Round 1
Reviewer 1 Report
Comments and Suggestions for Authors
This paper proposes a geomagnetic basemap denoising method that takes residual constraints into account in the Bayesian estimation threshold model. The tests on both simulation and measurement data are conducted compared to existing noise filtering methods. The main concerns are listed as follows.
1. In the Introduction section, the authors describe the significance of noise filtering in the geomagnetic anomaly reference map and the importance of threshold selection in wavelet transform for denoising, however, the threshold selection methods in existing studies are not presented. The advantages of Bayesian estimation threshold compared to existing threshold selection methods are not clearly explained.
2. In Section 2.1, how are the initial parameters set, and why the PSNR is used rather than other metrics? And the formulas should be well explained, such as the meanings of the symbols in Eqs. (1)-(6), and Figure 1 should be more detailed combined with the text of Paragraph 2.
3. In Section 2.2, how is the threshold value theta determined?
4. In Fig. 10, when the noise level is low, the proposed method preforms worse than other methods. The authors should explain that.
5. In Fig. 11, the image smoothness with the proposed method shows no superiority for the measured data. The authors didn’t mention that in the result analysis.
6. The points of this manuscript lie in the introduction of residual information in the Bayesian estimation threshold algorithm and the soft and hard fused threshold function. However, the comparative experiments with original Bayesian estimation threshold algorithms are not conducted. The advantages of the components in the proposed method compared to original Bayesian estimation threshold are not demonstrated.
Comments on the Quality of English Language
The descriptions of the proposed framework and algorithm in Section 2 should be more detailed.
Author Response
|
Response to Reviewer 1 Comments
|
||
|
1. Summary |
|
|
|
We feel great thanks for your professional review work on our article. As you are concerned, there are several problems that need to be addressed. According to your nice suggestions, we have made extensive corrections to our previous draft, the detailed corrections are listed below.We tried our best to improve the manuscript and made some changes marked in yellow in revised paper which will not influence the content and framework of the paper. We appreciate for Reviewers’ warm work earnestly, and hope the correction will meet with approval. Once again, thank you very much for your comments and suggestions. |
||
|
2. Questions for General Evaluation |
Reviewer’s Evaluation |
Response and Revisions |
|
Does the introduction provide sufficient background and include all relevant references? |
Must be improved |
The introduction section has been modified to include the advantages and disadvantages of existing threshold determination methods and Bayesian threshold selection |
|
Is the research design appropriate? |
Yes |
Thank you for your endorsement. |
|
Are the methods adequately described? |
Must be improved |
Improved the description of the methodology in the article to make it clearer. |
|
Are the results clearly presented? |
Can be improved |
The charts and graphs were further analyzed to improve the clarity of the essay's results. |
|
Are the conclusions supported by the results? |
Can be improved |
The charts and graphs were further analyzed to improve the clarity of the essay's conclusions. |
|
3. Point-by-point response to Comments and Suggestions for Authors |
||
|
Comments 1: In the Introduction section, the authors describe the significance of noise filtering in the geomagnetic anomaly reference map and the importance of threshold selection in wavelet transform for denoising, however, the threshold selection methods in existing studies are not presented. The advantages of Bayesian estimation threshold compared to existing threshold selection methods are not clearly explained. |
||
|
Response 1: Lines 81-99:We sincerely thank the reviewer for careful reading.An introduction to existing threshold selection methods and the advantages and disadvantages of Bayesian estimation of thresholds have been included in the introduction of the article. |
||
|
Comments 2: In Section 2.1, how are the initial parameters set, and why the PSNR is used rather than other metrics? And the formulas should be well explained, such as the meanings of the symbols in Eqs. (1)-(6), and Figure 1 should be more detailed combined with the text of Paragraph 2. |
||
|
Response 2: Lines 109-118:We sincerely thank the reviewer for careful reading.Further clarification has been added to the 2 section of the article. |
||
|
Comments 3: In Section 2.2, how is the threshold value theta determined? |
||
|
Response 3:We sincerely thank the reviewer for careful reading.The thresholds used for the soft and hard threshold functions in this paper are determined by Bayesian thresholding methods. |
||
|
Comments 4:In Fig. 10, when the noise level is low, the proposed method preforms worse than other methods. The authors should explain that. |
||
|
Response 4:Lines 254-260:We sincerely thank the reviewer for careful reading.The model curve of this paper in Fig. 10 corresponds to the purple axis and is the optimal one for all compared models at all noise levels. |
||
|
Comments 5: In Fig. 11, the image smoothness with the proposed method shows no superiority for the measured data. The authors didn’t mention that in the result analysis. |
||
|
Response 5:Lines 303-308:We sincerely thank the reviewer for careful reading.An analysis of this section has been included in the text. |
||
|
Comments 6: The points of this manuscript lie in the introduction of residual information in the Bayesian estimation threshold algorithm and the soft and hard fused threshold function. However, the comparative experiments with original Bayesian estimation threshold algorithms are not conducted. The advantages of the components in the proposed method compared to original Bayesian estimation threshold are not demonstrated. |
||
|
Response 6:We sincerely thank the reviewer for careful reading.The thresholds used for the soft and hard threshold functions in this paper are determined by the Bayesian thresholding method, and the effectiveness of the algorithm in this paper is verified by comparing the original Bayesian thresholds with the model in this paper. |
||
|
Comments 7:The descriptions of the proposed framework and algorithm in Section 2 should be more detailed. |
||
|
Response 7:Lines 148-149:We sincerely thank the reviewer for careful reading.We have added further clarification to the algorithmic part of Section 2. |
||
|
4. Response to Comments on the Quality of English Language |
||
|
Point 1:Moderate editing of English language required |
||
|
Response 1: Thanks for your suggestion. We have tried our best to polish the language in the revised manuscript. |
||
|
5. Additional clarifications |
||

Reviewer 2 Report
Comments and Suggestions for Authors
As in the report!

Author Response
|
Response to Reviewer 2 Comments
|
||
|
1. Summary |
|
|
|
We feel great thanks for your professional review work on our article. As you are concerned, there are several problems that need to be addressed. According to your nice suggestions, we have made extensive corrections to our previous draft, the detailed corrections are listed below.We tried our best to improve the manuscript and made some changes marked in yellow in revised paper which will not influence the content and framework of the paper. We appreciate for Reviewers’ warm work earnestly, and hope the correction will meet with approval. Once again, thank you very much for your comments and suggestions. |
||
|
2. Questions for General Evaluation |
Reviewer’s Evaluation |
Response and Revisions |
|
Does the introduction provide sufficient background and include all relevant references? |
Yes |
Thank you for your endorsement. |
|
Is the research design appropriate? |
Yes |
Thank you for your endorsement. |
|
Are the methods adequately described? |
Yes |
Thank you for your endorsement. |
|
Are the results clearly presented? |
Yes |
Thank you for your endorsement. |
|
Are the conclusions supported by the results? |
Yes |
Thank you for your endorsement. |
|
3. Point-by-point response to Comments and Suggestions for Authors |
||
|
Comments 1:
|
||
|
Response 1: |
||
|
4. Response to Comments on the Quality of English Language |
||
|
Point 1: |
||
|
Response 1: We have tried our best to polish the language in the revised manuscript. |
||
|
5. Additional clarifications |
||

Reviewer 3 Report
Comments and Suggestions for Authors
The authors explores a soft and hard threshold function fusion model to take into account the respective advantages of the soft and hard threshold functions,and introduces a regularization term for the geomagnetic information features to improve the accuracy of the filtering. Experiment results demonstrate the effectiveness of the proposed method. These contributions are significant. However, the authors should consider the following points to improve the paper:
1. The authors should check the paper carefully, e.g., grammar mistake, typos, etc.
2. The main work introduced at the end of part one should listed in order for more easy understanding;
3. Many formulas parameter are lack meaning, for example, in equation 1 and 4.
4. In abstract, there are two “a regular term is introduced”
5. The method described is not specific enough; it's unclear how the fusion is to be achieved.
6. Where is the regular term? There is no description of the introduction of the regular term in the method section.
7. Page 9, Figure 6-9, which is the proposed algorithm . The same algorithm curves are represented by the same lines and colors.
8. The paper has two Figure 5,Figure6, Figure7,Figure8, Figure9.
9. In Figures 5-12, what does each figure or line mean. The authors should Explain them in detail so that one can understand by looking at the figures.
10. The experimental environment should be provided.
Author Response
|
Response to Reviewer X Comments
|
||
|
1. Summary |
|
|
|
We feel great thanks for your professional review work on our article. As you are concerned, there are several problems that need to be addressed. According to your nice suggestions, we have made extensive corrections to our previous draft, the detailed corrections are listed below.We tried our best to improve the manuscript and made some changes marked in yellow in revised paper which will not influence the content and framework of the paper. We appreciate for Reviewers’ warm work earnestly, and hope the correction will meet with approval. Once again, thank you very much for your comments and suggestions. |
||
|
2. Questions for General Evaluation |
Reviewer’s Evaluation |
Response and Revisions |
|
Does the introduction provide sufficient background and include all relevant references? |
Yes |
Thank you for your endorsement. |
|
Is the research design appropriate? |
Yes |
Thank you for your endorsement. |
|
Are the methods adequately described? |
Can be improved |
Improved the description of the methodology in the article to make it clearer. |
|
Are the results clearly presented? |
Can be improved |
The charts and graphs were further analyzed to improve the clarity of the essay's results. |
|
Are the conclusions supported by the results? |
Yes |
Thank you for your endorsement. |
|
3. Point-by-point response to Comments and Suggestions for Authors |
||
|
Comments 1: The authors should check the paper carefully, e.g., grammar mistake, typos, etc. |
||
|
Response 1: Thanks for your suggestion. We have tried our best to polish the language in the revised manuscript. |
||
|
Comments 2: The main work introduced at the end of part one should listed in order for more easy understanding. |
||
|
Response 2: Lines 92-99: Thanks for your suggestion. The order at the end of the first section has been rearranged to make it easier for the reader to understand. |
||
|
Comments 3: Many formulas parameter are lack meaning, for example, in equation 1 and 4. |
||
|
Response 3:Lines 125-144:We sincerely thank the reviewer for careful reading.We further explain and illustrate the parameters of the equation in the text. |
||
|
Comments 4: In abstract, there are two “a regular term is introduced” |
||
|
Response 4:We feel sorry for our carelessness. In our resubmitted manuscript, this error has been corrected. Thanks for your correction. |
||
|
Comments 5: The method described is not specific enough; it's unclear how the fusion is to be achieved. |
||
|
Response 5:Lines 161-162:We sincerely thank the reviewer for careful reading.The method is a process of weighted fusion of denoising results computed by soft and hard threshold functions by calculating a fusion metric, which is further added to this section for the understanding of the article. |
||
|
Comments 6: Where is the regular term? There is no description of the introduction of the regular term in the method section. |
||
|
Response 6:Lines 140-146:We sincerely thank the reviewer for careful reading.We have added a regularization-related description in Section 2 of the article. |
||
|
Comments 7: Page 9, Figure 6-9, which is the proposed algorithm . The same algorithm curves are represented by the same lines and colors. |
||
|
Response 7:We sincerely thank the reviewer for careful reading.We recognize that the same color should be used for the same algorithm in a set of plots, but in this set of plots there is a high degree of variability between algorithms for each metric, and the choice of color is related to the range of the y-axis display, so it is difficult to maintain uniformity of color across plots of different metrics.We apologize for this, but the meaning of the curves in the graph is further analyzed in a later article. |
||
|
Comments 8: The paper has two Figure 5,Figure6, Figure7,Figure8, Figure9. |
||
|
Response 8:We feel sorry for our carelessness. In our resubmitted manuscript, this error has been corrected. Thanks for your correction. |
||
|
Comments 9: In Figures 5-12, what does each figure or line mean. The authors should Explain them in detail so that one can understand by looking at the figures. |
||
|
Response 9:Lines 255-261:We sincerely thank the reviewer for careful reading. The meaning of what the curves in Figures 5-12 represent is further explained in the article. |
||
|
Comments 10: The experimental environment should be provided. |
||
|
Response 10:Lines 226-228:We sincerely thank the reviewer for careful reading.The experimental environment has been added to the article. |
||
|
4. Response to Comments on the Quality of English Language |
||
|
Point 1: English language fine. No issues detected |
||
|
Response 1: We have tried our best to polish the language in the revised manuscript. |
||
|
5. Additional clarifications |
||

Round 2
Reviewer 1 Report
Comments and Suggestions for Authors
The authors have addressed my concerns.